# *APOE4* and infectious diseases jointly contribute to brain glucose hypometabolism, a biomarker of Alzheimer's pathology: New findings from the ADNI

**Aravind Lathika Rajendrakumar** **\***, **Konstantin G. Arbeev, Olivia Bagley, Matt Duan, Anatoliy I. Yashin, Svetlana Ukraintseva\*, for the Alzheimer's Disease Neuroimaging Initiative**¶

Biodemography of Aging Research Unit, Social Science Research Institute, Duke University, Durham, North Carolina, United States of America

¶ Membership of the Alzheimer's Disease Neuroimaging Initiative is listed in the Acknowledgments.
\* svetlana.ukraintseva@duke.edu (SU); arl75@duke.edu (ALR)

**Data Availability Statement:** The ADNI consortium determines access to the data used in our analysis

## Abstract

### Background

Impaired brain glucose metabolism is a preclinical feature of neurodegenerative diseases such as Alzheimer's disease (AD). Infections may promote AD-related pathology. Therefore, we investigated the interplay between infections and *APOE4*, a strong genetic risk factor for AD.

### Methods

We analyzed data on 1,509 participants in the Alzheimer's Disease Neuroimaging Initiative (ADNI) database using multivariate linear regression models. The outcomes were rank-normalized hypometabolic convergence index (HCI), statistical regions of interest (SROI) for AD, and mild cognitive impairment (MCI). Marginal mean estimates for infections, stratified by *APOE4* carrier status, were then computed.

### Results

Prior infections were associated with greater HCI [β = 0.15, 95% CI: 0.03, 0.27, p = 0.01]. The combined effects of infections and *APOE4* carriers on HCI levels were significantly greater than either variable alone. Among *APOE4* carriers, the estimated marginal mean was 0.62, rising to 0.77, with infections (p<0.001), indicating an interaction effect. Carriers with multiple infections showed greater hypometabolism (higher HCI), with an estimate of 0.44 (p = 0.01) compared to 0.11 (p = 0.08) for those with a single infection, revealing a dose-response relationship. The estimates for the association of infections with SROI AD and SROI MCI were β = -0.01 (p = 0.02) and β = -0.01 (p = 0.04), respectively.

and defines the data as a "Limited Data Set" under 45 CFR Part 164.514 (https://adni.loni.usc.edu/wp-content/themes/adni_2023/documents/ADNI_DSP_Policy.pdf). The authors have utilized this data exclusively for secondary analysis. However, all data can be publicly accessed through ADNI (https://adni.loni.usc.edu/data-samples/adni-data/#AccessData) upon request and can be used for replication, following the methods described in the paper. All researchers will have the same level of access as the authors.

**Funding:** This research was supported by the National Institute on Aging of the National Institutes of Health under Award Numbers R01AG076019 and R01AG062623. The content is solely the responsibility of the authors and does not necessarily represent the official views of the National Institutes of Health.

**Competing interests:** The authors declare no competing interests.

**Abbreviations:** AD, Alzheimer's disease; AIC, Akaike Information Criterion; APOE, Apolipoprotein E; Aβ, Amyloid βeta; FDG, 18F-fluorodeoxyglucose; GxE, Gene-environment interaction; GWAS, Genome-wide association studies; HSV, Herpes Simplex Virus; %INCMSE, Percent Increase in Mean Squared error; IQR, Interquartile Range; PET, Positron Emission Tomography; pTau, Phosphorylated Tau; SD, Standard Deviation; SNP, Single Nucleotide Polymorphism; SROI, Statistical Region of Interest; NECTIN2, Nectin Cell Adhesion Molecule 2 (gene); PUD, Peptic Ulcer Disease; UTI, Urinary Tract Infection.

## Conclusion

Our findings suggest that infections and *APOE4* jointly contribute to brain glucose hypometabolism and AD pathology, supporting a "multi-hit" mechanism in AD development.

## Introduction

Alzheimer's disease (AD) is a slowly developing neurodegenerative disorder that is clinically manifested as dementia [1]. The current figure for the AD burden in older adults in the United States is 6.7 million, and it is poised to rise to 13.8 million by 2060 [2]. The preclinical stage of AD can last many years without obvious signs of dementia [3]. It is crucial to better understand this preclinical stage to develop successful AD prevention [4]. Common preclinical features of AD include toxic protein depositions, neuronal apoptosis, and reduction in hippocampal volume (brain shrinkage), and brain glucose hypometabolism [5, 6]. The brain glucose hypometabolism is observed long before the occurrence of overt symptoms in AD and is partly due to mitochondrial dysfunction [7]. Measuring glucose utilization in the brain using positron emission tomography (PET) and 18F-fluorodeoxyglucose (FDG) allows for convenient examination of hypometabolic patterns in the brain [8]. Brain scans based on FDG PET can effectively detect around 90% of AD-specific metabolic patterns, such as those in the parieto-temporal, frontal, and posterior cingulate regions [9].

A large genetic component drives AD (60–80%), and the entire spectrum of the disease can develop over 15–25 years [10]. Genetic variations in the *APOE* gene could single-handedly account for a large part of the risk related to AD in old age [11]. On the other hand, addressing modifiable risk factors could reduce or delay up to 40% of dementia risk [12]. Therefore, by focusing on the modifiable risk factors, a substantial part of the AD burden could be alleviated at the population level [13]. Prevention of certain infections can reduce the risk of chronic diseases, including neurological deficits [14–16]. Accumulating evidence suggests that infections could be a significant risk factor for AD that may also facilitate the development of AD pathology at the preclinical stage, though the exact mechanism is unclear and might involve a direct detrimental impact of infection-related factors as well as indirect effects of compromised immunity [17–20].

The connection between infections and AD and related pathology may also be influenced by genetic factors [18, 21, 22]. There are also indications that infections can contribute to brain hypometabolism, one of the earliest features of AD pathology; however, research on this topic is scarce [23]. Here we explored how infectious diseases may influence brain glucose metabolism in presence and absence of *APOE4*, the strongest genetic risk factor for AD, in participants of the Alzheimer's Disease Neuroimaging Initiative (ADNI).

## Methods

### Study design and population

ADNI is a multi-center observational study that began in 2004 under the supervision of Michael W. Weiner. The study recruited individuals within the 55–90 years age range, and enrollment in this cohort occurs in different phases, with previous participants continuing to be in the study and new participants being recruited. The ADNI study only includes subtypes of amnestic MCI. A physician diagnoses MCI and AD based on the medical history, cognition

scales such as the Alzheimer's Disease Assessment Scale–Cognitive (ADAS-COG), the Mini-Mental State Examination (MMSE), and biomarkers.

To compare and gain knowledge about dementia, this database maintains and updates demographic, phenotypic, biomarker, and genetic data gathered from participants with normal cognition, AD, and other forms of cognitive impairment. The availability of such a wide variety of biomarkers provides sufficient information to learn about the evolution and pathology driving AD [24].

More details regarding the study design and objectives can be accessed here (https://adni.loni.usc.edu/study-design/). Broadly, ADNI seeks to integrate information from biomarkers, cognitive measures, and brain scans to improve AD diagnosis and treatment [25]. Brain scans were primarily collected to learn about the structural and metabolic functions of the brain, serving as a standard for differentiating the pathological changes seen in AD from those in normal aging [26].

## Predictors: Infections and *APOE4*

Prior infections were determined by combining information from medical history, baseline symptoms, initial health assessment, and adverse effects datasets. The details of the selected subset of infections included in the final dataset are illustrated in Fig 1 in S1 File. Medical history information was collected during the screening visit using a questionnaire. Non-harmonious disease names were uniformly labeled for analytical purposes. Duplicated participant information having the same infection and diagnosis date, as well as any infections lacking a diagnosis date, were subsequently excluded. Covariates such as age, sex, education, race, marriage status, and *APOE4* information were retrieved from the ADNIMERGE file.

The *APOE4* carrier status was identified from DNA extracted by Cogenics from a 3 ml aliquot of EDTA blood extracted from participants during their screening visit [27]. Anti-diabetic medications were extracted (**list provided in the S2 File**) using the *Anatomical Therapeutic Chemical (ATC)* classification system coding (https://www.who.int/tools/atc-ddd-toolkit/atc-classification). Information regarding smoking and alcohol usage was obtained from the medical history file. Finally, we retained infections that only preceded the HCI measurements.

## Outcomes: Brain glucose hypometabolism, AD, and MCI

Multiple PET scanners were used to capture brain images based on a standard protocol [28]. Measures were taken to correct for the related discrepancies [29]. The details regarding the PET scan and related protocols can be viewed elsewhere (https://adni.loni.usc.edu/methods/pet-analysis-method/pet-analysis/). The generated raw PET data are centrally stored at the Laboratory of Neuroimaging (LONI) at the USC Mark and Mary Stevens Neuroimaging and Informatics Institute of the University of California [30].

We retrieved the processed study outcomes from the BAIPETNMRCFDG dataset (https://adni.bitbucket.io/reference/baipetnmrc.html). The main outcome of interest was the hypometabolic convergence index (HCI), developed to reflect AD-specific hypometabolism across regions of the brain by computing voxel-wise z-scores from FDG-PET brain images. Higher HCI values correspond to lower levels of metabolism in the brain [31].

Additionally, we examined the associations for infections with statistical regions of interest (SROI) corresponding to AD and Mild Cognitive Impairment (MCI). SROI associations might provide additional insights into the cerebral metabolic rate for glucose (CMRgl) decline in these brain regions, helping to understand the disease-specific pathology they represent [32]. The Statistical Parametric Mapping (SPM) software was used to generate the HCI and

SROI scores [32, 33]. The work of Landau et al. provides further details on the generation and development of regions of interest in the ADNI cohort [34]. A decline in FDG-PET Region of Interest (ROI) values suggests a pathological brain damage and may contribute to the progression of dementia [33].

### Statistical analysis

R version 4.3.2 was used for the data linking and statistical analysis [35]. We analyzed the dataset with full covariate and outcome information, without performing any imputations. The *ggplot2* package was used to create variable distribution plots [36]. The leptokurtic HCI readings were normalized during the *RNomni* package [37]. Multivariate linear regression models were conducted for all specified outcomes separately. Age, education, and allele dosages of *APOE ε4* were analyzed as continuous variables. Infections, AD, and diabetes medication use were coded as a binary variable (yes or no). Marriage, smoking, and alcohol use were coded as Ever or Never. We explored models with a full set and a reduced set of covariates. The parsimonious model (the best explanatory model) was determined using the Akaike Information Criterion (AIC) in the *MuMin* package [38]. This package iteratively runs multiple models with varying terms, assigning weights to each model based on its AIC to determine the optimal set of variables. This approach effectively reduces the number of variables in the final model without imposing the constraints typically associated with multiple comparison corrections, such as the Bonferroni test. A two-sided p-value less than 0.05 was considered to support our hypothesis.

A *Random Forest-based* model was used to rank the significant variables according to their contributions to the best model [39]. For this, we specified the following hyperparameters; 5000 trees, the number of features for splitting corresponding to the square root of the total number of variables, a node size of 1, bootstrapping with replacement, and Mean Squared Error (MSE) as the splitting criterion. The effect modification for infections with HCI by *APOE4* carrier status was assessed by visualizing with the *rockchalk* package [40]. Marginal mean estimates were calculated to show the interaction effects for the infections across categories of *APOE4* and sex.

### Ethics considerations

The Institutional Review Board of Duke University Health System issued approval for this study (Protocol IDs Pro00109279 and Pro00105389). This publication includes only secondary analyses of existing data available from ADNI, and does not include identifiable human data. Written informed consent for ADNI participants was obtained by the ADNI in accordance with the local legislation and ADNI requirements. ADNI studies follow Good Clinical Practices guidelines, the Declaration of Helsinki, and United States regulations (U.S. 21 CFR Part 50 and Part 56).

## Results

### Participant characteristics

The final sample included information on 1,509 participants after data linking (**Fig 2 in S1 File**). As shown in Table 1, the average age among participants was 73.3 years, with an average education duration of 16.0 years (IQR 14.0–18.0). Over 96% of respondents reported being ever married, and 55.8% were males. There was a relatively lower representation of non-white individuals, totaling 116 (7.6%) in the sample. Percentage of individuals with a history of smoking and alcohol use was 27.1% and 3.3%, respectively. Of these, 215 individuals

**Table 1. Demographic and clinical characteristics of the study population.**

| variable | Mean/Median /Frequency | SD/IQR | Range |
|---|---|---|---|
| Age (Years) [#] | 73.3 | 7.2 | 55.0–91.4 |
| Male (%) | 843 (55.8%) | | |
| Education (Years) [#] | 16.0 | 14.0–18.0 | 4.0–20.0 |
| Marriage Status | | | |
| Ever | 1455 (96.4%) | | |
| Never | 54 (3.5%) | | |
| Race | | | |
| White | 1393 (92.3%) | | |
| Other | 116 (7.6%) | | |
| Smoking (Ever) | 409 (27.1%) | | |
| Alcohol (Ever) | 50 (3.3%) | | |
| Infections (Yes) | 215 (14.2%) | | |
| Time duration (Years) [#@] | 8.4 | 3.5–28.3 | 0.03–86.7 |
| HCI [#] | 12.59 | 8.4–19.3 | 2.3–55.2 |
| SROI AD | 1.15 | 0.08 | 0.8–1.38 |
| SROI MCI | 1.03 | 0.10 | 0.7–1.35 |
| APOE4 [$] | | | |
| 0 | 813 (53.8%) | | |
| 1 | 544 (36.0%) | | |
| 2 | 152 (10.0%) | | |
| Diabetes (Yes) | 43 (2.8%) | | |
| Cognitive Status | | | |
| CN | 333 (22.1%) | | |
| SMC | 106 (7.0%) | | |
| EMCI | 334 (22.1%) | | |
| LMCI | 458 (30.3%) | | |
| AD | 277 (18.3%) | | |

Note. Data are presented as mean ± standard deviation (SD) or percentage (%) for continuous and categorical variables, respectively;

[#]Variables with skewed distributions are presented as median and IQR.

[$] Frequencies in the analyzed sample.

[@]Time from Infection to HCI measurements.

accounting for 14.2% of the total sample size, reported having infections. The median interval between biomarker assessment and infections was 8.4 (IQR: 3.5–28.3). Median HCI was 12.59, and the IQR was 8.4–19.3.

Fig 1 shows the distributions of the original HCI and rank-normalized HCI and also a scatterplot of their relationship. For AD and MCI participants, the mean SROI values were 1.15 and 1.03, respectively. About 3% of participants with diabetes were on medication. As regards the cognitive status of participants in the sample, 333 (22.1%) were cognitively normal (CN), 277 (18.3%) were diagnosed with AD, 334 (22.1%) had Early Mild Cognitive Impairment (EMCI), 458 (30.3%) had Late Mild Cognitive Impairment (LMCI), 106 (7.0%) had Significant Memory Concern (SMC), and 1 (0.0%) did not have a diagnosis. Peptic ulcer disease (PUD) (n = 152), urinary tract infection (UTI) (n = 146), and pneumonia (n = 102) were the most frequent among the selected infections. Fig 3 in S1 File shows the difference in the distribution of HCI values for individuals with infections, AD, and APOE4. The median HCI value among

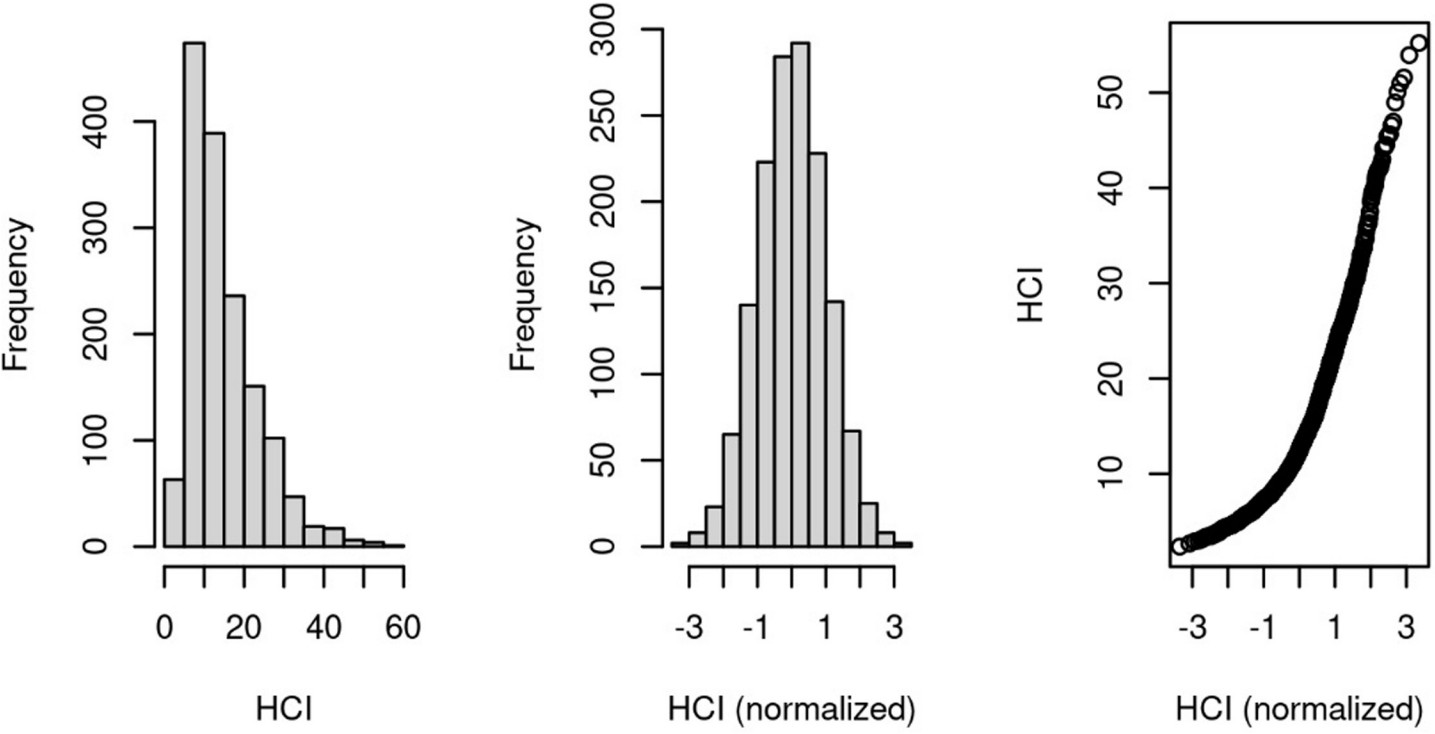

**Fig 1. Distribution of HCI and normalized HCI with scatterplot showing their relationship.**

individuals with infections was 13.64, while it was lower (12.48) for those without infections. It was also seen that the HCI had a modest positive correlation with *APOE4* (**Fig 4 in S1 File**).

## Association of infections and other predictors with the HCI

Table 1 in S1 File shows the regression estimates for all the predictors in the multivariate linear regression full model for HCI outcome. Marriage status, education, smoking, alcohol, and diabetes medication use were not significant predictors of HCI. Table 2 presents the results of the reduced model, which best describes the model variance. AD status predicted the strongest reduction in brain metabolism [β = 1.04, 95% CI 0.92–1.15, p<0.001], followed by age [β = 0.01,

**Table 2. Regression estimates for predictors in the reduced multivariate linear regression model for HCI outcome.**

| Variables | Estimates | 95% CI | P |
|---|---|---|---|
| AD (Yes) | 1.04 | 0.92, 1.15 | <0.001*** |
| *APOE4* | 0.32 | 0.25, 0.38 | <0.001*** |
| Age | 0.01 | 0.01, 0.02 | <0.001*** |
| Infections (Yes) | 0.15 | 0.02, 0.27 | 0.01* |
| Race (White) | 0.25 | 0.09, 0.42 | 0.002** |
| Sex (Male) | 0.17 | 0.09, 0.26 | <0.001*** |
| Smoking (Yes) | 0.08 | -0.01, 0.18 | 0.085 |

Note. *p<0.05;

**p<0.01;

***p<0.001.

95% CI 0.01–0.02, p<0.001] and *APOE4* carrier status [β = 0.32, 0.25–0.38, p<0.001]. Higher variable relevance is indicated by higher values of %INCMSE and INCNodepurity (**Table 2 in S1 File**). The regression coefficient for infections was 0.15 [95% CI 0.02–0.27, p = 0.01]. Males and white people were at higher risk of having elevated HCI values. Smoking history was the only non-significant predictor retained in the reduced model. The adjusted R-squared from the reduced model was 26.9%. Males had higher median HCI values.

In the sex-stratified analysis evaluating the effects of infections versus non-infections, males generally demonstrated relatively higher HCI values (**Table 3 in S1 File**). The difference in normalized marginal means between all groups was statistically significant (p<0.001). The combined effects of infections and *APOE4* carrier status on HCI levels are shown in Fig 2. This was significantly greater than the effects of either variable alone. Table 4 in S1 File clarifies these results. Specifically, for individuals without infections and *APOE4* carrier status, the estimated marginal mean was 0.03 (p = 0.53). However, this increased significantly to 0.18 (p<0.001) for *APOE4* non-carriers in the presence of infections. Notably, among *APOE4* carriers, the estimated marginal mean was substantially higher at 0.62, and this value rose to 0.77 with infections (p<0.001), confirming an interaction between the two factors.

This interaction was further demonstrated in the additional analysis (**Table 5 in S1 File and Fig 3**), indicating that carriers who experienced multiple infections exhibited greater hypometabolism. Among individuals with more than one prior infection (n = 23), the estimate was significantly higher at 0.44 (p = 0.01) compared to those with a single infection, which was 0.11 (p = 0.08), revealing a dose-response relationship.

### Association of infections and other predictors with the SROI AD

Table 6 in S1 File provides regression estimates for all the factors investigated for SROI AD. Marriage, race, smoking, and alcohol history were not significant predictors of AD-specific hypometabolism. In the reduced model shown in Table 3, AD was associated with increased region-specific hypometabolism (regression coefficient: -0.08, p<0.001). The use of diabetes medications was associated with decreased brain metabolism (-0.03, p = 0.02). Similar to previous regression, an increase in *APOE4* alleles was a strong risk factor for hypometabolism (-0.02, p<0.001). Male gender showed greater hypometabolism (-0.01, p<0.01). Although education was linked to a better metabolic pattern, this relationship was not profound. Age-specific decreases were not as notable as those observed in HCI (-0.003, p<0.001). While statistically significant, the effect estimate for previous infections was lower for AD (-0.01, p = 0.02). These variables collectively predicted 26.8% of the variance in SROI AD.

### Association of infections and other predictors with the SROI MCI

The results of the SROI MCI regression (full model) is presented in the Table 7 in S1 File. Generally, the estimates were closer to the SROI AD than HCI. Among the variables that best explained the model (Table 4), AD, *APOE4*, and diabetes medications had the largest effect estimates. Sex and use of diabetic medications had a marginally greater impact on the MCI region than on the AD region. However, race and education were not identified as significant predictors. Infections were associated with a -0.01 reduction in regional metabolism (p = 0.04). The percentage of variation explained by the model for the SROI MCI was also the highest (28.8%) of the three investigated outcomes.

### Discussion

Results of our study suggest that infections and *APOE4* can jointly significantly affect brain glucose metabolism, specifically promote hypometabolism, as measured by the increased

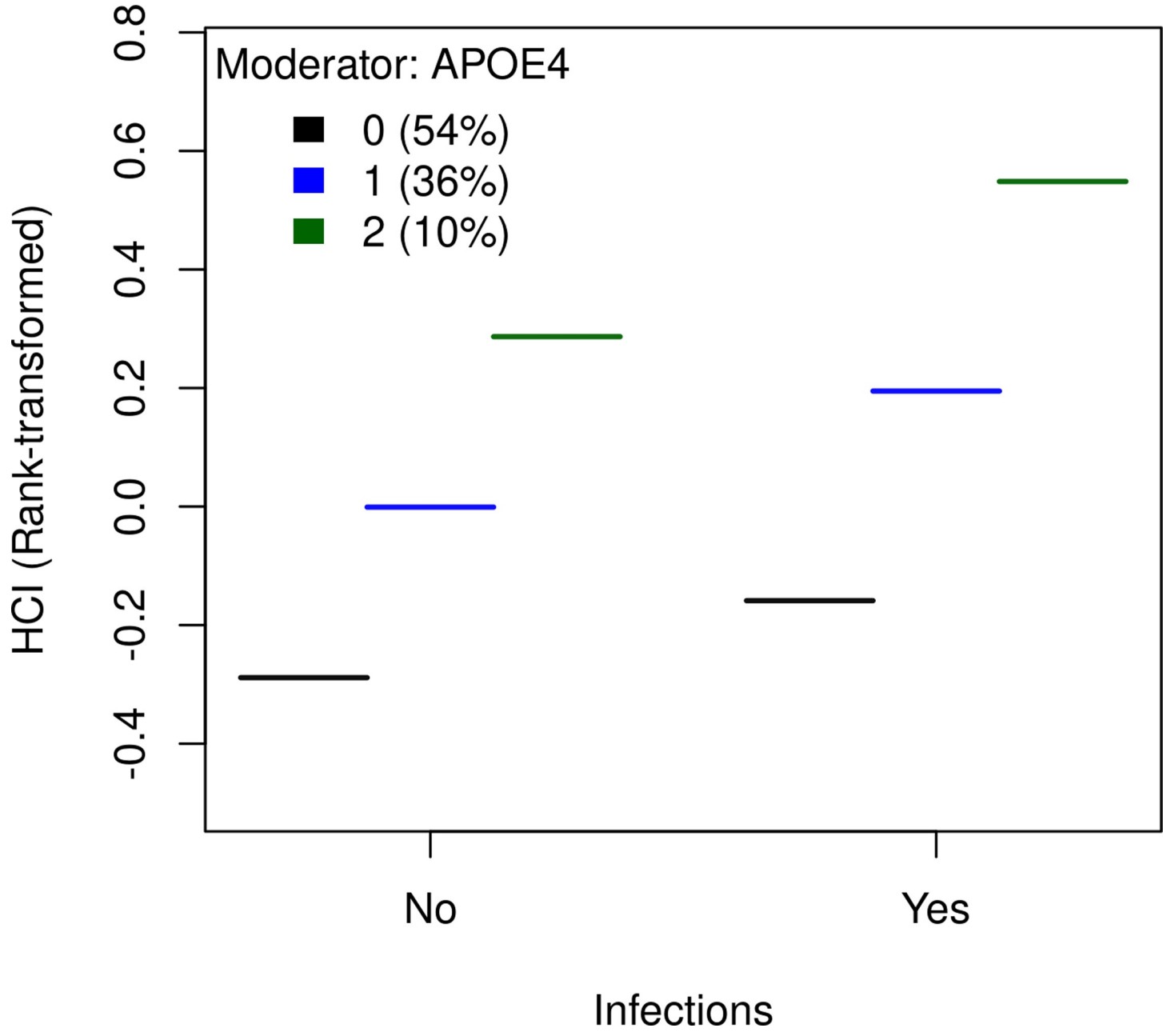

**Fig 2. Joint effect of *APOE4* and history of infections on HCI.**

values of HCI. A history of infections in this ADNI sample corresponds to a greater hypometabolism, specifically a 0.15 unit increase in rank normalized HCI. However, this estimate rose to 0.44 in the presence of multiple infections. Model inclusion of established confounders such as age, sex, race, and education did not diminish these findings. We also adjusted for AD status, which was more prevalent in the group with no infections and associated with reduced brain metabolism. Adjusting for AD status and *APOE4* was necessary to reveal the genuine association of previous infections. Additionally, previous infections were significantly associated with regional brain metabolism specific to AD and MCI in our

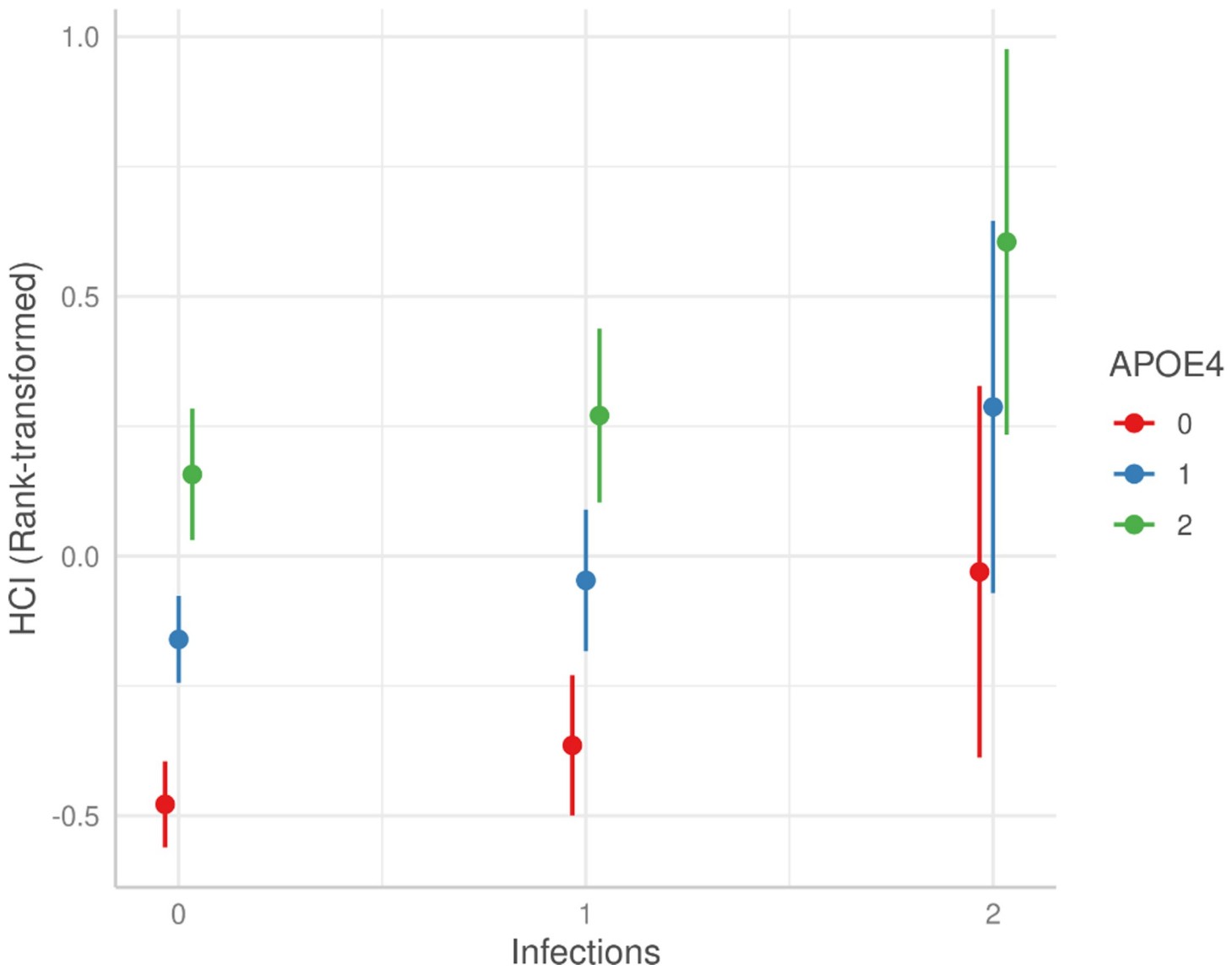

**Fig 3. Brain hypometabolism by *APOE4* carrier status and frequency of infections.**

data. The MCI cases are exclusive and not due to AD, as the final diagnosis status was considered in the analysis.

Our findings agree with previous research suggesting that infections may negatively impact brain metabolism [16, 41–43]. Infectious diseases, including those addressed in this study, have been previously linked to AD in other data [22, 44–46]. Our recent paper that used Health and Retirement Study (HRS) data reported associations between AD and various infectious (viral, bacterial, fungal), suggesting that compromised immunity may play a role in AD etiology [20]. The connection between infections and brain hypometabolism may also involve pathological immune responses. Some research provides indirect support to this idea by linking brain hypometabolism to microglia activation [47–49].

Determining the onset time of infection is a major challenge in AD research. Furthermore, the causal inferences are obscured by the fact that individuals with AD often grapple with a

**Table 3. Regression estimates for predictors in the reduced multivariate linear regression model for SROI AD outcome.**

| Variables | Estimates | 95% CI | P |
|---|---|---|---|
| AD (Yes) | -0.08 | -0.09, -0.07 | <0.001*** |
| *APOE4* | -0.02 | -0.03, -0.02 | <0.001*** |
| Age | -0.003 | -0.003, -0.002 | <0.001*** |
| Diabetes Medication | -0.03 | -0.05, -0.004 | 0.02* |
| Education | 0.001 | 0.00, 0.002 | 0.04* |
| Infections (Yes) | -0.01 | -0.02, -0.001 | 0.02* |
| Sex (Male) | -0.01 | -0.02, -0.003 | 0.00** |
| Smoking (Yes) | -0.01 | -0.02, 0.001 | 0.11 |

Note. *p<0.05;

**p<0.01;

***p<0.001.

variety of infections due to declining immunity, leading to elevated antimicrobial markers [16]. Pathogens have a high affinity to the central nervous system and brain tissue and could affect cognition [50]. Given that brain hypometabolism is an early sign of AD, our findings suggest that infections could potentially trigger this process [51]. However, the progression of hypometabolism may also depend upon the combination of other risk factors [52]. Infections can affect the brain through multiple pathways, both directly and indirectly, particularly when the blood-brain barrier is breached [53, 54]. Infections propagated through the respiratory route can also reach the brain relatively easily [50, 55]. However, upon reaching the brain, different infectious agents employ their preferred mechanisms, such as latent activation and the initiation of inflammation, as seen in the case of the Herpes virus [56]. Pathogen invasion into the brain leads to chronic inflammation, which can compromise the blood-brain barrier [42]. There are distinct differences in inflammatory pathways noted across specific pathogens [19, 57]. Age-related changes could exacerbate these pathological processes even further [58]. Strom and colleagues have also demonstrated that brain hypometabolism correlates with tau pathology and neurodegeneration in crucial dementia-specific regions within the ADNI dataset [59]. These mechanisms could potentially explain a significant portion of the biological processes leading up to hypometabolism.

**Table 4. Regression estimates for predictors in the reduced multivariate linear regression model for SROI MCI outcome.**

| Variables | Estimates | 95% CI | P |
|---|---|---|---|
| AD (Yes) | -0.09 | -0.10, -0.08 | <0.001*** |
| *APOE4* | -0.02 | -0.03, -0.02 | <0.001*** |
| Age | -0.004 | -0.01, -0.003 | <0.001*** |
| Diabetes Medication (Yes) | -0.04 | -0.06, -0.01 | 0.01** |
| Infections (Yes) | -0.01 | -0.03, -0.0005 | 0.04* |
| Sex (Male) | -0.02 | -0.03, -0.01 | <0.001*** |
| Smoking (Yes) | -0.01 | -0.02, 0.003 | 0.15 |

Note. *p<0.05;

**p<0.01;

***p<0.001.

Studies on the relationship between *APOE4* and brain metabolism have produced contrasting findings [59–63]. In their recently published work, Fortea and colleagues found that simply being homozygous for *APOE4* is sufficient, in most cases, to guarantee an AD diagnosis [64]. In our analysis, the increase in *APOE4* allele was associated with all three outcomes and showed compounding effects with infections and their burden. Even in patients with a single *APOE4* variant, which is usually not considered a significant increase in risk compared to homozygous carriers, the presence of infections increases the risk of hypometabolism to nearly the same level as in homozygous *APOE4* carriers. One possibility is that the observed effect is due to accelerated neuroinflammation arising from the presence of both risk factors [65]. Risk factors for AD tend to cluster in individuals with *APOE* risk alleles, including a reduction in brain metabolism [66]. Amyloid-beta and Tau deposition are higher in *APOE4* carriers [67]. *APOE4* can also accelerate brain degeneration through non-overlapping pathways independent of amyloid deposition and Tau pathology [68–70]. *APOE4* alleles both promote and resist infections, depending on the type of infection [71]. Researchers suggest that *APOE4* polymorphisms result in increased lipid production [72] and blood-brain barrier loss [73], which could facilitate a conducive environment for pathogens [74]. Supporting evidence from the Northern Manhattan Study showed that the effect-modifying relationship between *APOE4* and infectious burden was correlated with decreased cognition [75]. The influence of *APOE4* on AD remains incompletely understood, although it is known to engage in intricate interactions with other risk factors for AD, such as age [76, 77]. However, in stark contrast to these findings, a study reported that the effects of *APOE4* on cognition are AD-specific. It singles out the cause of cognitive decline as the interaction between *APOE4* and amyloid beta in the hippocampus [78].

Our study revealed that diabetes medication was the third-biggest risk factor for AD and MCI-specific brain metabolism, but not for the HCI measure. Previous studies indeed demonstrated that diabetes increases the risk for MCI and AD [79, 80]. Individuals with diabetes and AD often share common biological pathways [81]. Most prominent among these are low-grade chronic inflammation and insulin resistance [82].

Sex differences in mechanisms related to AD warrant in-depth study. Usually, females are susceptible to AD and early brain hypometabolism compared to males [83, 84]. On exposure to prior infections, women are also, particularly at higher risk for reduced hippocampal volume [19]. Males overall had a higher HCI value than females in our sample. It is important to note that males with infections had a slightly higher mean age. But this alone cannot explain the gender difference. Importantly, there was no difference in the increase in marginal means due to infections for both sexes. An earlier study reported that brain hypometabolism increased in men after 70 years of age, while this was not seen in females in a normal brain [85]. However, the applicability of this finding in the AD context needs confirmation. Some participant characteristics in ADNI may differ from the general population due to voluntary recruitment. Variations in the distribution of AD risk factors among genders might also contribute to this finding [86, 87].

Given that AD is not curable, prevention stands as the most viable option at present. Vaccinations may potentially alleviate dementia risk. Influenza vaccines, in particular, are among the candidates demonstrating this preventive potential [4, 79]. However, personal genetics could play a role in determining the efficacy and effectiveness of vaccinations. Recent research has revealed that individuals carrying a polymorphism in the *NECTIN2* gene exhibit a decreased susceptibility to AD when compared to non-carriers, when receiving vaccinations for pneumonia and flu [22].

The availability of medical history information and longitudinally standardized FDG PET measurements were important strengths of our study. We were also able to demonstrate the

temporality of association, which was rarely described in earlier human studies [88]. There were a couple of study limitations. Of these, the most important is that the medical history is questionnaire-based, suggesting that recall bias may exist and lead to an incorrect exposure classification. Currently, the representation of high-risk groups, such as Afro-American and Hispanic individuals, is limited in the ADNI database, which has constrained the evaluation of the effect modification role of race in relation to infections and brain hypometabolism [89]. In this work, we did not specifically explore the heterogeneity in infections and the brain metabolism relationship. However, previous AD studies indicate that there could be subgroups that may be differentially vulnerable [90–92]. The distribution of the time from the onset of infection and HCI measurement was highly variable in our sample. As individuals with more than one prior infection were limited in our dataset, we could not further investigate whether the timing of infection, along with number of previous infections, would accelerate brain hypometabolism. There may also be a cohort effect, wherein the frequency of infections observed within this group may not accurately fit the current disease landscape. We recommend validating the findings in large cohorts with robust information on prior infections.

## Conclusion

This study found that infections and *APOE4* jointly promoted brain glucose hypometabolism in older ADNI participants. In individuals with a history of infections who were also carriers of one *APOE4* allele, the degree of brain glucose hypometabolism was nearly that seen in *APOE4* homozygotes without prior infections. We conclude that prior infections may contribute to AD pathology in synergy with *APOE4*, thus playing a part in the "multi-hit" mechanism of AD development.

## Supporting information

**S1 File. Supplementary figures and tables.**
(DOCX)

**S2 File. List of anti-diabetes drugs.**
(XLSX)

## Acknowledgments

Data collection and sharing for this project was funded by the Alzheimer's Disease Neuroimaging Initiative (ADNI) (National Institutes of Health Grant U01 AG024904) and DOD ADNI (Department of Defense award number W81XWH-12-2-0012). ADNI is funded by the National Institute on Aging, the National Institute of Biomedical Imaging and Bioengineering, and through generous contributions from the following: AbbVie, Alzheimer's Association; Alzheimer's Drug Discovery Foundation; Araclon Biotech; BioClinica, Inc.; Biogen; Bristol-Myers Squibb Company; CereSpir, Inc.; Eisai Inc.; Elan Pharmaceuticals, Inc.; Eli Lilly and Company; EuroImmun; F. Hoffmann-La Roche Ltd and its affiliated company Genentech, Inc.; Fujirebio; GE Healthcare; IXICO Ltd.; Janssen Alzheimer Immunotherapy Research & Development, LLC.; Johnson & Johnson Pharmaceutical Research & Development LLC.; Lumosity; Lundbeck; Merck & Co., Inc.; Meso Scale Diagnostics, LLC.; NeuroRx Research; Neurotrack Technologies; Novartis Pharmaceuticals Corporation; Pfizer Inc.; Piramal Imaging; Servier; Takeda Pharmaceutical Company; and Transition Therapeutics. The Canadian Institutes of Health Research is providing funds to support ADNI clinical sites in Canada. Private sector contributions are facilitated by the Foundation for the National Institutes of Health (www.fnih.org). The grantee organization is the Northern California Institute for Research

and Education, and the study is coordinated by the Alzheimer's Disease Cooperative Study at the University of California, San Diego. ADNI data are disseminated by the Laboratory for Neuro Imaging at the University of Southern California.

Data used in preparation of this article were obtained from the Alzheimer's Disease Neuroimaging Initiative (ADNI) database (adni.loni.usc.edu). As such, the investigators within the ADNI contributed to the design and implementation of ADNI and/or provided data but did not participate in analysis or writing of this report. A complete listing of ADNI investigators can be found at: http://adni.loni.usc.edu/wp-content/uploads/how_to_apply/ADNI_Acknowledgement_List.pdf.

## Author Contributions

**Conceptualization:** Aravind Lathika Rajendrakumar, Konstantin G. Arbeev, Anatoliy I. Yashin, Svetlana Ukraintseva.

**Data curation:** Aravind Lathika Rajendrakumar, Olivia Bagley, Matt Duan.

**Formal analysis:** Aravind Lathika Rajendrakumar, Konstantin G. Arbeev, Anatoliy I. Yashin, Svetlana Ukraintseva.

**Funding acquisition:** Svetlana Ukraintseva.

**Investigation:** Konstantin G. Arbeev.

**Methodology:** Konstantin G. Arbeev.

**Project administration:** Svetlana Ukraintseva.

**Supervision:** Konstantin G. Arbeev, Anatoliy I. Yashin, Svetlana Ukraintseva.

**Writing – original draft:** Aravind Lathika Rajendrakumar.

**Writing – review & editing:** Aravind Lathika Rajendrakumar, Konstantin G. Arbeev, Anatoliy I. Yashin, Svetlana Ukraintseva.

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
