## [Decision Letter · Decision Letter 0]

29 Oct 2024

PONE-D-24-43325APOE4 and Infectious Diseases Jointly Contribute to Brain Glucose Hypometabolism, a Biomarker of Alzheimer’s Pathology: New Findings from the ADNIPLOS ONE

Dear Dr. Lathika Rajendrakumar,

Thank you for submitting your manuscript to PLOS ONE. After careful consideration, we feel that it has merit but does not fully meet PLOS ONE’s publication criteria as it currently stands. Therefore, we invite you to submit a revised version of the manuscript that addresses the points raised during the review process.

We look forward to receiving your revised manuscript.

Kind regards,

Yi Su, Ph.D

Academic Editor

PLOS ONE

Journal Requirements:

“This research was supported by the National Institute on Aging of the National Institutes of Health under Award Numbers R01AG076019 and R01AG062623. The content is solely the responsibility of the authors and does not necessarily represent the official views of the National Institutes of Health.”

“Data collection and sharing for this project was funded by the Alzheimer's Disease Neuroimaging Initiative (ADNI) (National Institutes of Health Grant U01 AG024904) and DOD ADNI (Department of Defense award number W81XWH-12-2-0012)”

“This research was supported by the National Institute on Aging of the National Institutes of Health under Award Numbers R01AG076019 and R01AG062623. The content is solely the responsibility of the authors and does not necessarily represent the official views of the National Institutes of Health”

Reviewers' comments:

Reviewer's Responses to Questions

**Comments to the Author**

1. Is the manuscript technically sound, and do the data support the conclusions?

Reviewer #1: Yes

Reviewer #2: Partly

2. Has the statistical analysis been performed appropriately and rigorously? 

Reviewer #1: Yes

Reviewer #2: I Don't Know

3. Have the authors made all data underlying the findings in their manuscript fully available?

Reviewer #1: Yes

Reviewer #2: Yes

4. Is the manuscript presented in an intelligible fashion and written in standard English?

Reviewer #1: Yes

Reviewer #2: Yes

5. Review Comments to the Author

Reviewer #1: This report use the widely available FDG-PET (relatively speaking and especially in the US) to examine the impact of the modifiable risk factor, infection, on AD. It also emphasizes the preclinical stage and the prevention.

those are all important in the era of prevention of AD

The analysis methodology used in this study is sound, and the statisical analysis is carried out with careful co

nsideration.

The results of this study suggest that infections and APOE4 can jointly significantly affect brain glucose

metabolism, specifically promote hypometabolism, as measured by the increased values of HCI. This finding is nov

el and is worth to be shared in the AD research community and medical field

some minor concerns

1) dosage APOE4 was treated as continus variable? This should be fine but it implies that you used 0 1 and 2?

2) random forest model ONLY used for ranking of significant variables?

3) double check 55.8% were males. This could be right for the smaller sub-cohort of the ADNI data for this repor

t. In general, female % is higher for ADNI

4) Table 1 should also list MCI info. As it is, only AD is listed (the last row)

Reviewer #2: The manuscript by Rajendrakumar, et al investigated the associations between brain glucose hypometabolism and APOE4 status, as well as prior infections. The authors found that APOE4 status and prior infections interactively associated with glucose hypometabolism in the brain. I have a few concerns on the clearness of hypothesis and data/methodology presentation, as follows:

1.The authors should clarify if the MCI patients selected from the ADNI database are amnestic MCI (due to AD?) or other types. Moreover, were the MCI and AD patients defined clinically or biologically? Recently, the NIA-AA published a biomarker-based research framework (please see PMIDs: 29653606 and 27371494) for AD, often referred to as the AT(N) framework to categorize research participants who may have AD or a related disorder. Do the MCI and AD patients in this study meet the AT(N) criteria? Please specify the number of subjects in each diagnostic group, namely CN, MCI, and AD.

2.It is unclear if the present study is a hypothesis-driven or data-driven study. Could the authors elaborate why there would be any associations between prior infection history and brain glucose hypometabolism (N)? Should we also expect that the identified associations are related to amyloid-beta (A) and tau (T) pathologies? Would the timing and number of prior infections be related to the associations?

3.Please specify how multiple comparison issues were considered/addressed.

4.Please specify how the random forest model was constructed, for instance, the number of decision trees in the forest, the number of features (e.g., square root) to consider when looking for the best split, the maximum depth of each tree, the minimum number of samples required to split an internal node, the minimum number of samples required to be at a leaf node, whether bootstrap samples or permutations are used when building trees, the function (e.g., Gini impurity or entropy) to measure the quality of a split, etc.

6. PLOS authors have the option to publish the peer review history of their article (what does this mean?). If published, this will include your full peer review and any attached files.

Reviewer #1: No

Reviewer #2: No

---

## [Author Response · Author response to Decision Letter 0]

9 Dec 2024

Dear Editor,

Thank you for considering our manuscript for peer review in the PLOS One journal. We have amended our manuscript and highlighted the changes in the text in yellow. We sincerely hope for a positive decision from the journal. 

1.Please include the following items when submitting your revised manuscript: A rebuttal letter that responds to each point raised by the academic editor and reviewer(s). You should upload this letter as a separate file labeled 'Response to Reviewers'. A marked-up copy of your manuscript that highlights changes made to the original version. You should upload this as a separate file labeled 'Revised Manuscript with Track Changes'. An unmarked version of your revised paper without tracked changes. You should upload this as a separate file labeled 'Manuscript'. 

As suggested, we have made point-by-point responses to the queries raised during the review process. We have also renamed and uploaded the files.

2.When submitting your revision, we need you to address these additional requirements. 1. Please ensure that your manuscript meets PLOS ONE's style requirements, including those for file naming. The PLOS ONE style templates can be found at https://journals.plos.org/plosone/s/file?id=wjVg/PLOSOne_formatting_sample_main_body.pdf and https://journals.plos.org/plosone/s/file?id=ba62/PLOSOne_formatting_sample_title_authors_affiliations.pdf. 

We have updated our manuscript to suit PLOS One’s style requirements. Specifically, we have updated the file names, font of the headings, references, and figure names.

3.Thank you for stating in your Funding Statement: “This research was supported by the National Institute on Aging of the National Institutes of Health under Award Numbers R01AG076019 and R01AG062623. The content is solely the responsibility of the authors and does not necessarily represent the official views of the National Institutes of Health.” Please provide an amended statement that declares *all* the funding or sources of support (whether external or internal to your organization) received during this study, as detailed online in our guide for authors at http://journals.plos.org/plosone/s/submit-now. Please also include the statement “There was no additional external funding received for this study.” in your updated Funding Statement. Please include your amended Funding Statement within your cover letter. We will change the online submission form on your behalf. 

We have updated the text in the Funding section to meet the journal's requirements and also included it in the cover letter.

4.Thank you for stating the following in the Acknowledgments Section of your manuscript:

“Data collection and sharing for this project was funded by the Alzheimer's Disease Neuroimaging Initiative (ADNI) (National Institutes of Health Grant U01 AG024904) and DOD ADNI (Department of Defense award number W81XWH-12-2-0012)”

We note that you have provided additional information within the Acknowledgements Section that is not currently declared in your Funding Statement. Please note that funding information should not appear in the Acknowledgments section or other areas of your manuscript. We will only publish funding information present in the Funding Statement section of the online submission form. Please remove any funding-related text from the manuscript and let us know how you would like to update your Funding Statement. Currently, your Funding Statement reads as follows:

“This research was supported by the National Institute on Aging of the National Institutes of Health under Award Numbers R01AG076019 and R01AG062623. The content is solely the responsibility of the authors and does not necessarily represent the official views of the National Institutes of Health”

The ADNI Data Publication Committee (DPC) requires that the following text be included in the Acknowledgment section, and that the consortium name be listed along with the authors, marked with an asterisk symbol. We have also noted that similar text has been published in PLOS One articles that used ADNI data (PMID: 38635499, PMID: 32716914). Therefore, we respectfully request that the editor include this text in order to comply with ADNI requirements.

We have updated the text under the Funding section within the cover letter.

5.We note that you have indicated that there are restrictions to data sharing for this study. For studies involving human research participant data or other sensitive data, we encourage authors to share de-identified or anonymized data. However, when data cannot be publicly shared for ethical reasons, we allow authors to make their data sets available upon request. For information on unacceptable data access restrictions, please see http://journals.plos.org/plosone/s/data-availability#loc-unacceptable-data-access-restrictions. Before we proceed with your manuscript, please address the following prompts: a) If there are ethical or legal restrictions on sharing a de-identified data set, please explain them in detail (e.g., data contain potentially identifying or sensitive patient information, data are owned by a third-party organization, etc.) and who has imposed them (e.g., a Research Ethics Committee or Institutional Review Board, etc.). Please also provide contact information for a data access committee, ethics committee, or other institutional body to which data requests may be sent. b) If there are no restrictions, please upload the minimal anonymized data set necessary to replicate your study findings to a stable, public repository and provide us with the relevant URLs, DOIs, or accession numbers. Please see http://www.bmj.com/content/340/bmj.c181.long for guidelines on how to de-identify and prepare clinical data for publication. For a list of recommended repositories, please see https://journals.plos.org/plosone/s/recommended repositories. You also have the option of uploading the data as Supporting Information files, but we would recommend depositing data directly to a data repository if possible. Please update your Data Availability statement in the submission form accordingly 

The ADNI consortium determines access to the data used in our analysis. The authors have utilized this data exclusively for secondary data analysis. ADNI policy (https://adni.loni.usc.edu/wp-content/themes/adni_2023/documents/ADNI_DSP_Policy.pdf) defines the data as a “Limited Data Set” under 45 CFR Part 164.514. Furthermore, the ADNI Data Use Agreement (https://adni.loni.usc.edu/wp-content/themes/adni_2023/documents/ADNI_Data_Use_Agreement.pdf), specifically item 3, prohibits the sharing of individual-level data. Therefore, we are unable to share the data used in this study or make it available upon request. However, all the data can be accessed through ADNI (https://adni.loni.usc.edu/data-samples/adni-data/#AccessData) and can be used for replication following the methods described in the manuscript. All researchers will have the same level of access as the authors.

We have added captions to our supporting information files and related in-text citations to match the description.

7.While revising your submission, please upload your figure files to the Preflight Analysis and Conversion Engine (PACE) digital diagnostic tool, https://pacev2.apexcovantage.com/. PACE helps ensure that figures meet PLOS requirements. To use PACE, you must first register as a user. Registration is free. Then, login and navigate to the UPLOAD tab, where you will find detailed instructions on how to use the tool. If you encounter any issues or have any questions when using PACE, please email PLOS at figures@plos.org. Please note that Supporting Information files do not need this step. 

We have uploaded modified images provided by the PACE tool.

Reviewer’s comments

Reviewer 1 

1.Have the authors made all data underlying the findings in their manuscript fully available? The PLOS Data policy requires authors to make all data underlying the findings described in their manuscript fully available without restriction, with rare exception (please refer to the Data Availability Statement in the manuscript PDF file). The data should be provided as part of the manuscript or its supporting information, or deposited to a public repository. For example, in addition to summary statistics, the data points behind means, medians and variance measures should be available. If there are restrictions on publicly sharing data—e.g. participant privacy or use of data from a third party—those must be specified. 

The ADNI consortium determines access to the data used in our analysis. The authors have utilized this data exclusively for secondary data analysis. ADNI policy (https://adni.loni.usc.edu/wp-content/themes/adni_2023/documents/ADNI_DSP_Policy.pdf) defines the data as a “Limited Data Set” under 45 CFR Part 164.514. Furthermore, the ADNI Data use agreement (https://adni.loni.usc.edu/wp-content/themes/adni_2023/documents/ADNI_Data_Use_Agreement.pdf), specifically item 3, does not permit the sharing of individual-level data. Therefore, we are unable to share the data used in this study or make it available upon request. However, all the data can be accessed through ADNI (https://adni.loni.usc.edu/data-samples/adni-data/#AccessData) and can be used for replication following the methods described in the manuscript. All researchers will have the same level of access as the authors.

2.Some minor concerns

dosage APOE4 was treated as a continuous variable? This should be fine but it implies that you used 0 1 and 2? 

We modeled the dosage of APOE4 as a continuous variable in Table 2. However, this approach does not allow for a detailed examination of how the effect of infections on brain hypometabolism varies between APOE4 single and double carriers. Therefore, in Figures 2 and 3, we stratified the effects by APOE4 status: 0, 1, and 2.

3. random forest model only used for ranking of significant variables? 

Yes, we have only used the random forest model to rank significant variables.

4. double check 55.8% were males. This could be right for the smaller sub-cohort of the ADNI data for this report. In general, female % is higher for ADNI 

We agree with the reviewer’s observation. As rightly pointed out, our analysis was conducted in a linked dataset. Therefore, this would likely differ from the characteristics of the entire cohort.

5. Table 1 should also list MCI info. As it is, only AD is listed (the last row) 

We have included the frequencies of cognitively normal, MCI, and SMC in Table 1.

Reviewer 2 

1. The manuscript by Rajendrakumar, et al investigated the associations between brain glucose hypometabolism and APOE4 status, as well as prior infections. The authors found that APOE4 status and prior infections interactively associated with glucose hypometabolism in the brain. I have a few concerns on the clearness of hypothesis and data/methodology presentation, as follows:The authors should clarify if the MCI patients selected from the ADNI database are amnestic MCI (due to AD?) or other types. Moreover, were the MCI and AD patients defined clinically or biologically? Recently, the NIA-AA published a biomarker-based research framework (please see PMIDs: 29653606 and 27371494) for AD, often referred to as the AT(N) framework to categorize research participants who may have AD or a related disorder. Do the MCI and AD patients in this study meet the AT(N) criteria? 

Please specify the number of subjects in each diagnostic group, namely CN, MCI, and AD. 

Indeed, we agree with the reviewer that more clarity is required for the readers regarding the representation of cognitive issues and the definition of MCI and AD in the ADNI sample we analyzed.We used the clinical diagnosis for participants at each visit. The HCI values were extracted for the latest visit and diagnosis. The MCI cases are exclusive and not due to AD, as the final diagnosis status was considered in the analysis. The ADNI study protocol only includes subtypes of amnestic MCI. We have added these points to the manuscript (lines: 85-87, 270-271)

We want to thank the reviewer for suggesting the articles. A physician diagnoses MCI and AD based on the medical history, cognition scales such as the Alzheimer’s Disease Assessment Scale–Cognitive (ADAS-COG), the Mini-Mental State Examination (MMSE), and biomarkers. While it is important to acknowledge the evolving criterion of AD diagnosis, a large part of the samples in the analysis were collected before the publication of new guidelines. Also, the readings suggest that the ATN criteria were devised for research purposes rather than replacing the clinical diagnosis. We have now included the frequencies of cognitively normal, MCI, and SMC in lines 172-175 and Table 1.

2. It is unclear if the present study is a hypothesis-driven or data-driven study. Could the authors elaborate why there would be any associations between prior infection history and brain glucose hypometabolism (N)? Should we also expect that the identified associations are related to amyloid-beta (A) and tau (T) pathologies? Would the timing and number of prior infections be related to the associations? 

We thank the reviewer for raising these important questions. Our study was conducted to complement previous research linking infections to cognitive disorders. For example, prior studies have shown that infections can increase the risk of dementia and Alzheimer’s disease (AD). Hence the present study is a hypothesis-driven study. However, information regarding the mechanisms driving this association remains scarce. Our study addresses this gap by demonstrating that infections impact a physiological process that is a defining feature of cognitive disorders. Notably, previous studies have elicited an association between infections with amyloid buildup and CSF ptau-181 and other phosphorylation isoforms (PMID: 38438446, PMID: 30001512, PMID: 38924651). We have previously demonstrated a connection between NECTIN2 gene polymorphism (known to increase susceptibility to infection) and elevated pTau-181 levels in the ADNI data (PMID: 39165837).The distribution of the time from the onset of infection and HCI measurement was highly variable in our sample. As individuals with more than one prior infection were limited in our dataset, we could not further investigate whether the timing of infection, along with number of previous infections, would accelerate brain hypometabolism. We have added the above lines in the limitation part (lines 347-350).

3. Please specify how multiple comparison issues were considered/addressed. 

To select the optimal variable combinations for our model, we used functions from the MuMin package in R. This package iteratively runs multiple models with varying terms, assigning weights to each model based on its AIC to determine the optimal set of variables. This approach effectively reduces the number of variables in the final model without imposing the constraints typically associated with multiple comparison corrections, such as the Bonferroni test (lines 140-143).

4. PLOS authors have the option to publish the peer review history of their article (what does this mean?). If published, this will include your full peer review and any attached files. Yes, we agree to publish the peer review history.

5. Please specify how the random forest model was constructed, for instance, the number of decision trees in the forest, the number of features (e.g., square root) to consider when looking for the best split, the maximum depth of each tree, the minimum number of samples required to split an internal node, the minimum number of samples required to be at a leaf node, whether bootstrap samples or permutations are used when building trees, the function (

---

## [Decision Letter · Decision Letter 1]

18 Dec 2024

APOE4 and Infectious Diseases Jointly Contribute to Brain Glucose Hypometabolism, a Biomarker of Alzheimer’s Pathology: New Findings from the ADNI

PONE-D-24-43325R1

Dear Dr. Lathika Rajendrakumar,

We’re pleased to inform you that your manuscript has been judged scientifically suitable for publication and will be formally accepted for publication once it meets all outstanding technical requirements.

Kind regards,

Yi Su, Ph.D

Academic Editor

PLOS ONE

Additional Editor Comments (optional):

Reviewers' comments:

Reviewer's Responses to Questions

**Comments to the Author**

1. If the authors have adequately addressed your comments raised in a previous round of review and you feel that this manuscript is now acceptable for publication, you may indicate that here to bypass the “Comments to the Author” section, enter your conflict of interest statement in the “Confidential to Editor” section, and submit your "Accept" recommendation.

Reviewer #1: All comments have been addressed

Reviewer #2: All comments have been addressed

2. Is the manuscript technically sound, and do the data support the conclusions?

Reviewer #1: Yes

Reviewer #2: Yes

3. Has the statistical analysis been performed appropriately and rigorously? 

Reviewer #1: Yes

Reviewer #2: Yes

4. Have the authors made all data underlying the findings in their manuscript fully available?

Reviewer #1: Yes

Reviewer #2: Yes

5. Is the manuscript presented in an intelligible fashion and written in standard English?

Reviewer #1: Yes

Reviewer #2: Yes

6. Review Comments to the Author

Reviewer #1: response from the authors is satisfactory, addressed all the concerns. not more suggested changes from this reviewer.

Reviewer #2: The authors have addressed my concerns. I therefore do not have any further comments. Congratulations!

7. PLOS authors have the option to publish the peer review history of their article (what does this mean?). If published, this will include your full peer review and any attached files.

Reviewer #1: **Yes: **Kewei Chen

Reviewer #2: No

---

## [Editor Report · Acceptance letter]

23 Dec 2024

PONE-D-24-43325R1 

PLOS ONE

Dear Dr. Lathika Rajendrakumar, 

I'm pleased to inform you that your manuscript has been deemed suitable for publication in PLOS ONE. Congratulations! Your manuscript is now being handed over to our production team.

Kind regards, 

on behalf of

Dr. Yi Su 

Academic Editor

PLOS ONE